# *AtRAC7/ROP9* Small GTPase Regulates *A. thaliana* Immune Systems in Response to *B. cinerea* Infection

**DOI:** 10.3390/ijms25010591

**Published:** 2024-01-02

**Authors:** Ivette García-Soto, Damien Formey, Angélica Mora-Toledo, Luis Cárdenas, Wendy Aragón, Alexandre Tromas, Arianna Duque-Ortiz, Juan Francisco Jiménez-Bremont, Mario Serrano

**Affiliations:** 1Centro de Ciencias Genómicas, Universidad Nacional Autónoma de México, Cuernavaca 62210, Morelos, Mexico; formey@ccg.unam.mx (D.F.); angie_mt@ciencias.unam.mx (A.M.-T.); 2Programa de Doctorado en Ciencias Bioquímicas, Centro de Ciencias Genómicas, Universidad Nacional Autónoma de México, Cuernavaca 62210, Morelos, Mexico; 3Facultad de Ciencias, Universidad Nacional Autónoma de México, Coyoacan 04510, Ciudad de México, Mexico; 4Instituto de Biotecnología, Universidad Nacional Autónoma de México, Cuernavaca 62210, Morelos, Mexico; luis.cardenas@ibt.unam.mx; 5Instituto de Biociencias, Universidad Autónoma de Chiapas, Blvd. Príncipe Akishino s/n, Tapachula 30798, Chiapas, Mexico; waragon@ccg.unam.mx; 6La Cité College, Bureau de la Recherche et de l’Innovation, Ottawa, ON K1K 4R3, Canada; atroma@lacitec.on.ca; 7Instituto Potosino de Investigación Científica y Tecnológica (IPICYT), San Luis Potosí 78216, San Luis Potosí, Mexico; arianna.duque@ipicyt.edu.mx (A.D.-O.); jbremont@ipicyt.edu.mx (J.F.J.-B.)

**Keywords:** *B. cinerea*, *RAC7*, actin cytoskeleton, immunity, transcriptional regulation

## Abstract

*Botrytis cinerea* is a necrotrophic fungus that can cause gray mold in over 1400 plant species. Once it is detected by *Arabidopsis thaliana*, several defense responses are activated against this fungus. The proper activation of these defenses determines plant susceptibility or resistance. It has been proposed that the RAC/ROP small GTPases might serve as a molecular link in this process. In this study, we investigate the potential role of the Arabidopsis *RAC7* gene during infection with *B. cinerea*. For that, we evaluated *A. thaliana RAC7-OX* lines, characterized by the overexpression of the *RAC7* gene. Our results reveal that these *RAC7-OX* lines displayed increased susceptibility to *B. cinerea* infection, with enhanced fungal colonization and earlier lesion development. Additionally, they exhibited heightened sensitivity to bacterial infections caused by *Pseudomonas syringae* and *Pectobacterium brasiliense*. By characterizing plant canonical defense mechanisms and performing transcriptomic profiling, we determined that *RAC7-OX* lines impaired the plant transcriptomic response before and during *B. cinerea* infection. Global pathway analysis of differentially expressed genes suggested that *RAC7* influences pathogen perception, cell wall homeostasis, signal transduction, and biosynthesis and response to hormones and antimicrobial compounds through actin filament modulation. Herein, we pointed out, for first time, the negative role of *RAC7* small GTPase during *A. thaliana*–*B. cinerea* interaction.

## 1. Introduction

Plant pathogenic fungi cause between 70 and 80% of plant diseases [1]. *Botrytis cinerea* is a necrotrophic fungus that can infect more than 1400 different plant species causing significant pre- and post-harvest losses with an estimated cost of up to $100 billion dollars per year [2,3]. This fungus causes gray mold disease and is considered the second most important phytopathogen worldwide [3,4]. The infection mechanism of *B. cinerea* includes several developmental stages such as penetration of the host surface, killing of host tissue with primary lesion formation, tissue maceration due to lesion expansion, and sporulation [5]. During the penetration of the host’s surface, *B. cinerea* uses cell-wall-degrading enzymes and, to a lesser extent, penetration structures such as appressorium [6]. This fungus also self-produces reactive oxygen species (ROS) and induces plant accumulation of ROS to phytotoxic levels to kill plant cells [5].

Although *B. cinerea* is an adapted phytopathogen, it requires a convergence of environmental, microbial and plant factors to successfully infect the plant [7]. Plants have a wide variety of physical, chemical, and genetic barriers to prevent pathogen invasion and development. The proper activation of these defenses determines plant susceptibility or resistance [8]. Plant physical barriers include the cuticle and cell wall, which prevent pathogen entry. However, pathogens can bypass these barriers and penetrate plant tissues through natural openings such as stomata and hydatodes or through injuries. Plants can stop pathogen invasion by activating the innate immune system, including Pattern-Triggered Immunity (PTI) and Effector-Triggered Immunity (ETI). These systems are based on the early detection of consensus molecules from microorganisms called microbe- or pathogen-associated molecular patterns (MAMPs or PAMPs). MAMPs are detected by the plants through pattern recognition receptors (PRRs) located in the plasma membrane [9]. Once PRRs recognize MAMPs, several signaling pathways are triggered in the cell. These pathways induce an increase in Ca^++^ flux, cytoskeleton re-arrangement, activation of Mitogen-Activated Protein Kinases (MAPKs), ROS production, cytoplasmatic alkalinization, callose deposition, and the increased accumulation of antimicrobial products. The goal of all these coordinated actions is the prevention of pathogen entry and progression [10]. However, some pathogens can evade the PTI response through effectors that suppress the recognition of MAMPs by PRRs or the signaling triggered by such recognition. Effectors can be recognized by intracellular receptors (R proteins), inducing the ETI. The ETI response is generally stronger than the PTI, triggering the hypersensitive response (HR). HR is recognized by visible lesions on infected tissue [11,12]. After the induction of PTI and ETI, the late defense responses are regulated by phytohormones such as salicylic acid (SA), jasmonic acid (JA), and ethylene (ET). These phytohormones play important roles in signal transduction and defense responses. However, other phytohormones such as auxin, abscisic acid, cytokinins, and brassinosteroids also modulate signaling networks involved in the immune process [8,11]. Although plant immune responses are well described, there are large gaps in our knowledge about signal transduction pathways from PAMP/MAMP perception to induction of late defense responses.

RAC/ROP small GTPases could be important players in the plant immunity signaling pathway because of their known function as molecular switches. The role of RACs/ROPs in pathogenesis has been studied in *Oryza sativa* (rice), *Hordeum vulgare* (barley), *Nicotiana tabacum* (tobacco) and *Arabidopsis thaliana*. Several RAC/ROP small GTPases have been silenced to determine their role in the immune response in rice [13]. The *Osrac5* and *Osrac4* mutants were less susceptible to *Magnaporthe grisea*, while the *Osrac1* mutant increased susceptibility to the pathogen. These results suggest that RACs/ROPs may play a dual role in the immune response [14].

Among the rice RACs/ROPs, *OsRAC1* has been the most studied. Several reports [13,14,15] have proposed that overexpression of constitutively active (CA) mutants of RACs/ROPs, modifies the interaction between *O. sativa* and *M. grisea*. For instance, it has been proposed that *OsRAC1* could be a regulator of ROS production by interacting with the N-terminus of the NADPH oxidase RBOHB [16]. Another pathway in which *OsRAC1* might participate during pathogen attack is the activation of MAPK signaling, but the mechanism is still unclear. These results suggest that *OsRAC1* is involved in the response to this pathogen through multiple pathways. There are 11 RAC/ROP members in *A. thaliana* [17], of which *AtRAC7/ROP9* is the orthologous gene to *OsRAC1* (https://www.arabidopsis.org/ (accessed on 25 February 2020)). However, to our knowledge, there is no information about its role in the *A. thaliana* immune response. Here, we demonstrate, for the first time, that *RAC7* is involved in the *A. thaliana* immune response against *B. cinerea* (and probably other pathogens) notably by transcriptional downregulation of several defense-related pathways.

## 2. Results

### 2.1. AtRAC7/ROP9 Overexpression Enhances Susceptibility to Different Pathogens

To identify the orthologs of rice RACs/ROPs in *A. thaliana*, specifically focusing on the *OsRAC1* gene, we performed a phylogenetic analysis (Appendix A). Based on this analysis, we determined that *AtRAC7*/*ROP9* (hereinafter referred to as *RAC7*) is the orthologous gene of *OsRAC1* (77% identity), which has a clear role in the defense of *O. sativa* against *M. grisea*.

Based on this, we wondered whether the *A. thaliana RAC7* gene plays a role in responding to pathogenic microorganisms. Previous works have described that RAC/ROPs proteins in *A. thaliana* are highly conserved (Appendix A), and that there is no phenotypic effect when the corresponding loss-of-function mutants are studied, hence constitutively active (CA) and overexpressing mutant lines were used to characterize their function [17].

To investigate the possible role of *RAC7*, we analyzed two *RAC7* T-DNA insertion lines (SALK_019272.56.00.x, *RAC7.1-OX*; SALK_015223.56.00.x, *RAC7.2-OX*). Upon obtaining homozygous lines, we determined the expression level of *RAC7*. To our surprise, we observed that indeed both T-DNA insertion lines overexpress *RAC7* between 6- and 13-fold more than Col-0 plants (Figure 1A).

To determine whether increased *RAC7* gene expression influences the plant–microbe interactions, we proceeded to challenge both T-DNA insertional lines of *RAC7*, which exhibit overexpression of *RAC7* (hereafter referred to as *RAC7.1-OX* and *RAC7.2-OX*) with different pathogens. *RAC7.1-OX* and *RAC7.2-OX* lines, and Col-0 plants were infected with *B. cinerea*, *Pseudomonas syringae* DC3000, and *Pectobacterium brasiliense* BF45. At 72 h post infection (HPI) with *B. cinerea*, we observed a 100% incidence and larger lesion area in both *RAC7-OX* lines, which were a two-fold larger lesion area as compared Col-0 (Figure 1B). Furthermore, we observed that *RAC7-OX* lines were susceptible to both bacterial pathogens at 72 HPI (Figure 1C,D). These findings indicate that *Arabidopsis* lines exhibiting increased *RAC7* expression are more susceptible to pathogens with different lifestyles, implying that the dysregulation of this gene impacts the defense response against microbes. Since the fungus *B. cinerea* is a highly significant phytopathogenic organism known to infect a wide range of plant species, in the following sections we will specifically concentrate on characterizing the *RAC7-OX* lines in response to this pathogen.

In barley, overexpression of type II CA-HvRAC/ROPs has been reported to increase penetration by *Blumeria graminis* [18]. To assess whether the susceptibility of *RAC7-OX* lines is due to changes in early *B. cinerea* infection, we performed a time course experiment to follow lesion development at 40, 48 and 72 HPI. The lesion caused by *B. cinerea* starts to be visible at 48 HPI, but in fact it was already visible at 40 HPI in *RAC7-OX* lines, but not in Col-0 plants (Figure 2A).

To correlate early infection with *B. cinerea* in *RAC7-OX* lines with increased infection, we quantified *B. cinerea* in infected tissue by qPCR as previously described [19]. We found that *B. cinerea* increased at a similar rate compared to Col-0 plants at 6 and 16 HPI, but at 40 HPI there were more pathogens infecting *RAC7-OX* lines than Col-0 plants, a behavior that was maintained until 72 HPI (Figure 2B). These observations were also confirmed by observing *RAC7-OX* lines and Col-0 leaves colonized by *B. cinerea* expressing GFP, in which case more hyphae were observed growing in *RAC7-OX* lines (Figure 2C). These results suggest that *RAC7* overexpression increases the susceptibility of *A. thaliana* by allowing an increase in *B. cinerea* infection.

### 2.2. RAC7 Expression Is Suppressed during B. cinerea Infection

We evaluated the transcriptional response of *RAC7* in leaves infected with *B. cinerea* to understand RAC7 behavior in response to this fungus. In Figure 3, we observe reduced expression of the *RAC7* gene in Col-0 plants after infection with *B. cinerea* compared to uninfected Col-0 plants. This downregulation of *RAC7* expression was more pronounced at 40 h post-fungal inoculation than at 6 h. This result, combined with the increased susceptibility of the plants upon *RAC7* overexpression, suggests that *RAC7* transcriptional reduction is necessary to activate the defense against *B. cinerea*.

RAC proteins undergo post-translational lipid modifications that allow them to be anchored to the cytoplasmic membrane and thus activated. In the case of RAC type II like RAC7, they are S-acylated and thus anchored to the membrane and re-localize to the cytosol when they are not S-acylated [20,21]. Therefore, we analyzed at the protein level the behavior of RAC7 (35s:GFP:RAC7) in *N. benthamiana* in the presence and absence of *B. cinerea*. We observed that under basal conditions RAC7 is mainly located in the cytoplasmic membrane, but 24 h after infection we observed a higher signal intensity in the cytosol and a decrease in the membrane (Figure 3B). These results suggest that RAC7 protein could change its subcellular location in response to *B. cinerea* from the cytoplasmatic membrane at cytosol (probably by induced de-acylation), causing its inactivation.

### 2.3. Canonical Plant Defense Mechanisms Are Not Disrupted by RAC7 Overexpression

Plants have performed chemical and physical barriers that attempt to prevent pathogen development such as cuticle, cell wall, ROS accumulation, and callose deposition [22,23,24,25]. To assess whether *RAC7-OX* lines were impacted in canonical plant defense mechanisms, we evaluated ROS accumulation, and callose deposition at 0, 6, 16, and 40 HPI (Figure 4A,B). In addition, we measured permeability by quantifying chlorophyll leaching for 5 h, at intervals of 1 h (Figure 4C). We found that the levels of ROS, callose and chlorophyll leaching were similar in *RAC7-OX* lines compared to Col-0 plants (Figure 4A–C). These results suggest that the susceptibility caused by *RAC7* overexpression is not due to these changes in canonical defense mechanisms.

### 2.4. Actin Cytoskeleton Architecture Is Affected in RAC7-OX Lines Prior to Interaction with B. cinerea

ROP GTPases have been reported to interact with and regulate regulatory elements of actin filaments which play an important role in the plant defense response to pathogens [20,26].

Therefore, in this study, we investigated the arrangement of actin filaments in Col-0 and *RAC7-OX* plants. To compare the arrangement of actin filaments in Col-0 and *RAC7-OX*, phalloidin staining was performed on plants treated with the pathogen (6 HPI) and compared to untreated samples (Figure 5). The observations indicate that under basal conditions, Col-0 plants depict the normal thick, integral, and longitudinally distributed actin filaments in epidermal cells (Figure 5A). Conversely, *RAC7-OX* lines exhibit actin filaments that are more disorganized, more numerous and shorter in length (Figure 5B,C). In the presence of the pathogen, Col-0 plants exhibited a clear disorganization, with fewer long filaments and a greater number of shorter filaments (Figure 5D), this actin organization resembles to that observed in *RAC7-OX* plants before pathogen exposure. Meanwhile, *RAC7-OX* plants treated with *B. cinerea* showed no differences compared to their basal state or non-treated condition, except for thinner filaments and greater filament fragmentation (Figure 5C,F). Previously, it has been demonstrated that plants with elevated actin abundance and treated with the actin polymerization inhibitor, LatB, display a higher susceptibility to *P. syringae* [27]. Therefore, it is plausible to hypothesize that *RAC7-OX* plants are also more susceptible to pathogens due to increased actin abundance and reduced filament polymerization, even in the absence of a pathogen infection. Therefore, we decided to investigate the transcriptomic response of *RAC7-OX* to inspect the status of genes related to pathogen response.

### 2.5. The Transcriptome of the RAC7-OX Line Is Modified before Interaction with the Pathogen Takes Place

To determine the molecular basis behind the differential response of the *RAC7-OX* line to *B. cinerea* and understand why *RAC7* overexpressors were susceptible to this pathogen, we compared the transcriptomic profiles of *RAC7.2-OX* at 0 and 40 HPI with those of the Col-0. We selected *RAC7.2-OX* (hereinafter referred to as *RAC7-OX*, unless otherwise clarified), based on the level of expression (Figure 1A). Interestingly, we observed that at basal conditions (0 HPI) *RAC7-OX* had many differentially expressed genes (DEGs) with 8425 (4403 downregulated and 4022 upregulated), while at 40 HPI only 251 DEGs were identified (88 downregulated and 163 upregulated) (Figure 6A).

These data indicate that *RAC7* overexpression induces a differential transcriptomic response even before the plant is infected with *B. cinerea*, and the number of DEGs decreases as the infection progresses. In addition, we found many DEGs that were unique to each studied time point (Figure 6B). These results indicate that *RAC7* overexpression alters the plant transcriptome even under unchallenged conditions.

### 2.6. Overexpression of RAC7 Disrupts Regulation of Genes Involved in Development and Plant Defense

To identify the functional category of down and up DEGs, we performed GO enrichment analysis. We found several enriched categories related to developmental and defense processes (Figure 7).

RAC/ROPs proteins are involved in several plant functions such as polar and diffuse growth, vesicle trafficking, signaling during biological processes, and pathogen response, among others [28]. We observed several categories related to plant defense, such as biological processes involved in interspecies interaction between organisms, cell death, cell-wall organization or biogenesis, cellular response to hormone stimulus, immune system processes, positive regulation of reactive oxygen species’ biosynthesis processes, regulation of defense response, response to bacteria, and signaling (Figure 7). Almost all of these categories were downregulated at 0 HPI and remained down at 40 HPI. These results indicate that *RAC7* overexpression impairs the expression of basal defense genes and prevents the proper activation of the plant immune response.

On the other hand, since *RAC7* has been reported as a player in embryo and root system development [29], we evaluated different phenotypic parameters to confirm whether the *RAC7* overexpression affects plant development. However, when exanimated, seed germination, pollen and pavement cell development, as well as rosette and root growth, did not display significant differences compared to Col-0 plants (Appendix A). This suggests that *RAC7* overexpression did not have a noticeable impact on plant growth and development.

### 2.7. Pathways Related to Pathogen Response Are Suppressed in RAC7-OX Line

To further characterize the pathogen response-related DEGs identified in the *RAC7-OX line*, we examined the changes in the regulation of several plant defense-related pathways over time (0 and 40 HPI) (Figure 8). Using MapMan representation, we observed that genes related to the cell wall, PR proteins, signaling, TFs, hormone metabolism, jasmonate metabolism, and secondary metabolism, such as glucosinolate biosynthesis, were differentially expressed at the basal condition and almost all of them were negatively regulated at this time (Figure 8A, Appendix A).

Although almost all of these pathways were less represented at 40 HPI, they remained downregulated, especially the signaling pathway (Figure 8B, Appendix A). These results suggest that RAC7 is involved in *A. thaliana* immunity during *B. cinerea* infection. However, when this gene is overexpressed in the *RAC7.2-OX* line, it induces transcriptomic changes, even in the absence of pathogen inoculation. This, in turn, results in the downregulation of essential plant defense pathways, spanning from pathogen perception to the induction of plant defense genes.

### 2.8. Exogenous Application of Hormones and Secondary Metabolites Partially Reverts the RAC7-Induced Susceptibility

As the hormonal balance is very important for plant defense against pathogens [30], we focused on the better-characterized plant defense-related pathways against *B. cinerea*, such as those involving JA, SA and glucosinolates. For each biosynthetic pathway, we analyzed the expression of the principal JA-, SA- and glucosinolate-induced genes (Figure 9).

In the case of the JA- and SA-related pathways, both were downregulated at 0 HPI (Figure 9A,B). Regarding the secondary metabolism, we observed that the glucosinolate-related pathway was downregulated at 0 HPI, with no change at 40 HPI (Figure 9C). These results agree with the susceptibility phenotype observed in the *RAC7-OX* lines during fungal infection. Thus, *RAC7*-overexpressing plants are more predisposed to *B. cinerea* infection due to the hormonal balance deficit. Although the hormonal transcriptomic profile at 40 HPI in the *RAC7-OX* line resembled that of the Col-0, it does not seem to be enough to stop *B. cinerea* infection.

We observed that under basal conditions (control), the genes *LOX2*, *PDF1.2*, and *PR1*, related to the response mediated by JA and SA, respectively, were downregulated in *RAC7-OX* plants (Figure 9D), whereas the genes *ICS1* and *NPR1* showed no differential expression in *RAC7-OX* with respect to Col-0. These results, as we observed in the RNAseq data, confirm that these hormone response marker genes are repressed prior to pathogen entry into the plant.

On the other hand, from all the analyzed genes, *LOX2* was downregulated at all conditions, regardless of the pathogen applied, suggesting that the expression of this gene is independent of infection. Moreover, in the presence of *B. cinerea* (40 HPI), only *LOX2* and *NPR1* genes were downregulated in both *RAC7-OX* lines, as observed in the RNAseq data (Figure 9A,B). However, in the presence of *P. brasilense* (24 HPI), most of the tested genes were downregulated in the *RAC7-OX* lines with respect to Col-0, whereas, in the presence of *P. syringae* infection (24 HPI), we did not observe a differential response of the tested genes, except for *LOX2* and *NPR1* only for one of the insertional lines. These results suggest that the genes analyzed are more responsive to infection by necrotrophic pathogens than by biotrophic pathogens. Furthermore, to validate our transcriptomic analysis and investigate the potential involvement of the JA, SA, and glucosinolate pathways in the susceptibility of *RAC7-OX* to *B. cinerea*, we supplemented the *RAC7* overexpression line with exogenous MeJA, SA, and camalexin and assessed the response against *B. cinerea*. We observed that the *RAC7* overexpression line was able to reactivate its defense response against *B. cinerea* when treated with MeJA, but not when treated with SA (Figure 10A,B). However, we observed a reduction in lesion area in *RAC7-OX* plants treated with camalexin (Figure 10C). These results point to the notion that *RAC7-OX*-induced susceptibility can be restored when plants are treated with JA and camalexin.

## 3. Discussion

The role of RAC GTPases in plant–organism interactions was established more than 25 years ago [28], but the mechanisms are still unknown. RAC GTPases are molecular switches that are involved in signal transduction, linking external stimuli to cellular responses. This function is crucial for plants, which are sessile organisms in constant interaction with biotic and abiotic stimuli. RAC GTPases are involved in both mutualistic and negative interaction between plants and organisms. In *L. japonicus*, it has been demonstrated that *ROP3* and *ROP6* are involved in the early stage of rhizobia colonization [31,32]. On the other hand, RAC GTPases’ role in pathogenesis has been proposed due to its enhanced resistance or susceptibility when constitutive active (CA) or negative dominant (DN) mutants have been studied [14,33].

The most extensively studied roles of RACs in immune response are those found in *O. sativa* and *Hordeum vulgare*. *OsRAC1* has been described to be involved in pathogenesis as a positive regulator of PTI and ETI against *M. oryza*, by studying both the CA and DN versions [33,34]. Furthermore, in barley, *HvRACB*-silenced or -overexpressed plants showed an opposite establishment of *B. graminis* infection structures, suggesting that this protein is involved in the early invasion stage of the pathogen [35].

In *A. thaliana*, there are 11 *ROPs* genes, but elucidating the function of each of them has been a major challenge due to their high sequence conservation (72.22–98.28%) (Appendix A) and ubiquitous expression pattern, resulting in functional redundancy among their members. The alternative proposed to overcome these difficulties has been the study of ROP-overexpressing or gain-of-function plants [36]. This strategy has been used by several authors to determine the role of some *ROPs* during cell division, polar growth, morphogenesis, and plant–pathogen interaction. This is the case with [37], who studied the role of *AtROP6* against *Golovinomyces orontii* and *Blumeria graminis* by studying gain-of-function mutants. In the current study, we decided to investigate the role in the *A. thaliana–B. cinerea* interaction of two *RAC7* overexpression lines (*RAC7-OXs*) available on the Salk website to overcome the possible functional redundancy. Nevertheless, is important to mention that, for *RAC7*, we analyzed all the lines available in the Salk collection and none of them were truly knockdown mutants, indicating that apparently they are unviable in *A. thaliana*.

Here, we report that *RAC7* overexpression enhances the susceptibility of *A. thaliana* to several pathogens including *B. cinerea*, *P. syringae*, and *P. brasiliense*. Other RAC/ROP overexpressed lines have been associated with enhanced susceptibility; for example, in barley, *HvRAC3*, and *HvROP6* overexpression lines have also been described as being more susceptible to *B. graminis* [38]. Similarly, overexpression of *OsRACB* increases the susceptibility of *O. sativa* to *M. grisea* [39]. All of these overexpressing lines, as well as the *RAC7-OX* lines, exhibited a higher level of fungal infection in the plant tissue. This could suggest that some RACs could assist fungus penetration, through plant defense mechanism suppression or by supporting fungus penetration structures. It has been proposed that small GTPases could modulate the immune response through the modification of ROS levels, biosynthesis of cell wall components, modulation of signaling transduction, recycling of receptors, transport of defense precursors, and cytoskeleton rearrangement [28,33]. For example, the overexpression of *OsRAC1-CA* increases ROS accumulation and stops rice blast fungus development [40]. Furthermore, OsRAC1 interacts with CCR1 to induce monolignol biosynthesis and strengthen the cell wall [17,41]. On the other hand, it has been reported that RACs could modulate the immunity response by relocating the nucleus in the penetration site of the pathogen as RACB against *Hordeum vulgare.*

The cytoskeleton plays an important role in this process, and it has also been reported that it can support the penetration of pathogens into plant tissues by assisting the invasion of fungal penetration structures [42,43]. Nevertheless, it seems that not all RACs take part in the pathogenesis via the same mechanisms.

To elucidate which pathways were differentially regulated in the *RAC7-OX* lines, we performed RNAseq analysis at basal conditions and 40 h post-infection. A wide range of genes were affected in *RAC7-OX* at all of the studied times, but this decreased at 40 HPI, probably due to the necrosis events triggered by *B. cinerea*. Our results suggest that *RAC7* overexpression changed the plants’ transcriptomic profile at basal conditions even before the plants were challenged with *B. cinerea*. Furthermore, *HvRACB* overexpression was reported to induce changes in global gene expression at basal and inoculated conditions [44] similar to our results. Several studies have proposed that RACs can affect gene expression, mainly through their interaction with MAPK and RACK1 [45,46,47], but in mammals, Rho GTPase has been proposed to regulate gene expression by regulating actin filament rearrangement [48]. In *A. thaliana*, SPK1 promiscuously activates ROP proteins to regulate actin polymerization through the WAVE-ARP2/3 complex [49]. It has been hypothesized that actin could regulate the immune response by forming a regulatory complex with TFs and chromatin in the nucleus [49]. Nevertheless, the mechanism by which RAC/ROP might regulate gene expression is still unknown and is likely to be indirect. Especially as RAC proteins are located upstream on the signaling transduction pathways, their activity must impact several biological processes.

Indeed, the *RAC7-OX* line showed many differentially regulated gene ontology categories. Correct plant development and nutritional status are very important for proper plant defense [30]; in our results, we observed that the *RAC7-OX* line exhibited upregulated genes related to developmental processes (Figure 7), probably as a mechanism to overcome immune attenuation. However, we did not observe any difference in the development and growth of *RAC7-OX* lines (Appendix A). In terms of the regulation of immune response affected by *RAC7* overexpression, we observed several DEGs in *RAC7-OX* enriched in GO related to immune response (Figure 7). Most of these categories were downregulated, suggesting that *RAC7-OX* line is immunosuppressed prior to pathogen attack and is unable to induce a defense response after *B. cinerea* infection. This provides stronger evidence that the overexpression of the *RAC7* gene can have a negative impact on the immune response in *A. thaliana*, while seemingly not affecting developmental processes.

Although there are several references showing that RAC proteins influence the immune response, the main pathways or mechanisms are still not understood. Here, we show that several pathways related to plant defense were downregulated in the *RAC7-OX* line compared to Col-0 at basal condition and 40 HPI. These pathways included receptors, R proteins, hormone response, cell wall, signaling, PR proteins, and the secondary metabolism (Figure 8). These findings suggest that *RAC7* overexpression results in a dysregulated response that could impact various stages of plant defense, from pathogen perception to the defense response. Indeed, since most of the defense pathways were downregulated under basal conditions, we hypothesize that *RAC7* overexpression induces a general immunosuppression. This is further supported by the increased susceptibility to other pathogens, both necrotrophic (*P. brasiliense*) and hemibiotrophic (*P. syringae*), observed in the *RAC7-OX* overexpressing lines.

When pathogens land on the plant surface, there are several preformed plant structures as well as chemical barriers to prevent pathogen entry [48,50]. These barriers include the cuticle and the cell wall [51]. The necrotrophic pathogens, such as *B. cinerea* and *P. brasiliense*, have specialized cutinases to degrade the cuticle, once this barrier is overcome, the cell wall reacts. The integrity of the cell wall is disrupted by pathogen infection, which induces changes in the cell wall status, which activates plant defenses [52]. This cell wall status depends on the proper composition and structure of the wall. For example, some mutants affected in cellulose, pectin, xylene, lignin, or hemicellulose biosynthesis showed differential resistance to *B. cinerea* [52]. For example, *gae1* and *gae6* knockdown mutants, which are impaired in pectin content, showed enhanced susceptibility to *P. syringae* and *B. cinerea* [53]. On the other hand, *cesa 3/4/7/8* single mutants were resistant to *B. cinerea* and showed induction of hormone signaling and PR genes [54]. Pectin esterification is also important for strengthening the cell wall at the fungal penetration site. Herein, we observed that several genes involved in biosynthesis and the remodeling of cell wall, including *CESA 3/4/7/8*, *GAE1* and *GAE6*, were downregulated, pointing to the *RAC7-OX* line being impaired in cell wall homeostasis under basal conditions (Appendix A). This could influence *RAC7-OX* line susceptibility by disrupting the primary barrier when pathogens land on the plant surface. Moreover, 40 h after *B. cinerea* challenge, genes related to cell wall organization or biogenesis were induced in *RAC7-OX*. The genes related to this category were implied as belonging to expansins, pectate lyase, arabino galacturonases, and xyloglucosyl transferases biosynthesis, all of which are involved in cell wall relaxing (Appendix A). This points to the *RAC7-OX* cell wall being weakened before and during *B. cinerea* attack. Plants also induce biochemical mechanisms to strengthen the cell wall at the site of pathogen invasion [25]. For example, plants could induce lignin and callose deposition to stop pathogen penetration, but despite the cell wall biosynthesis disruption, the *RAC7-OX* line does not induce differential callose deposition during *B. cinerea* infection (Figure 5B).

Besides physical barriers such as the cell wall, correct and early pathogen perception is crucial for the activation of the plant immune response [55]. This perception is driven by the PRR on the plasma membrane, which senses PAMPs such as chitin from fungi [55]. In rice, the chitin receptor CERK1 is important for detection of *M. grisea*, and in *A. thaliana* it is required for chitin-induced resistance to *B. cinerea* [56]. This receptor is downregulated under basal conditions in the *RAC7-OX line*, which could point to a lack of *B. cinerea* perception. Other receptors such as RLP23, RLP43, RLP33, RLP39, RLP19, RLP18, RLP28, RLP50, RLP46, RLP49, RLP34, and RLP41 were also downregulated at 0 HPI and 40 HPI in the *RAC7-OX* line (Appendix A). The *rlp23* mutants are more susceptible than Col-0 to *B. cinerea*, because they are required to perceive BcNEP1 and BcNEP2, which are required during pathogen invasion and late infection, respectively [57]. The triggering of PTI by RLP23 requires the complex EDS1-PAD4-ADR1, this complex is also recruited by R proteins turning on ETI [58]. In the *RAC7-OX* line, the genes that code for the *EDS1-PAD4-ADR1* complex are downregulated, as are several R proteins (Appendix A). All of this suggests that *RAC7-OX* could be unable to sense *B. cinerea*, invasion and infection, thereby preventing PTI and ETI responses.

Once PRRs sense PAMPs, the intracellular Ca^++^ influx increases, triggering very important physiological responses such as protein kinase activation, ROS accumulation, actin rearrangement, gene expression, etc. These events take place very quickly after pathogen detection, and continue for several hours [59]. It has been proposed that Ca^++^ influx and ROS accumulation start between 1 and 30 min after elicitor treatment and have a cyclic peak during pathogen invasion [59]. The *RAC7-OX* line also showed downregulation of genes activated downstream of the Ca^++^ influx such as several calcium-dependent protein kinases (CDPK) (Appendix A), suggesting that calcium-dependent signaling transduction is also suppressed in the *RAC7-OX* line. However, we did not observe any difference in ROS accumulation in *RAC7-OX* compared to Col-0 (Figure 5A). The Ca^++^ are closely linked to cytoskeletal rearrangement. In *adf4* mutants, the loss of cytoskeletal dynamics in response to bacterial elicitors led to downregulation of CDPKs and pathogen response genes as in our overexpressed line, pointing to a possible regulation of the immune response by *RAC7* through the cytoskeleton- and Ca^++^-mediated signaling.

Other signaling pathways were also suppressed in these *RAC7* overexpressing lines, such as MAPK-dependent signaling pathways and phosphoinositide signaling (Appendix A). RACs are molecular switches involved in molecular signaling in various cellular processes. It has been reported that several RACs could interact with receptors in protein–protein or GEF-mediated interactions to transduce external signals [60,61] In this context, it has been proposed that OsRAC1 is triggered downstream of CERK1 via GEF1. Once activated, OsRAC1 interacts with multiple proteins to induce ROS accumulation, MAPK cascade, lignin deposition, and PTI response [33]. Overexpression of *RAC7* could suppress signaling through multiple pathways and it will be interesting to know its interactors during pathogen attack.

Downstream of signaling activation, plants trigger defense gene expression. Several of these genes, including PR proteins, hormone response, cell wall remodeling, and secondary metabolites with antimicrobial activity were downregulated in the *RAC7-OX* line (Figure 8, Appendix A). Plant hormonal response is very important in abolishing pathogen entry and propagation, specifically salicylic acid, ethylene, and jasmonic acid have been well studied in their roles in response to *B. cinereal* [62]. The role of SA in *B. cinerea* resistance is not clear, mutants in the SA pathway have not shown enhanced resistance or susceptibility to *B. cinerea* [8]. However, it has been well demonstrated that SA is required to limit fungal growth at the site of infection [8]. Based on this idea, it is possible that *B. cinerea* could infect the *RAC7-OX* line more efficiently due to the suppression of the SA pathway. However, despite our observation of SA pathway repression in the *RAC7-OX* line, the exogenous application of SA did not demonstrate a reduction in susceptibility. This further strengthens the notion that SA may not be involved in systemic resistance before infection by *B. cinerea* but could potentially impede the entry of the fungus.

On the other hand, ethylene and jasmonate-dependent responses are crucial for resistance to *B. cinerea*, and several mutants deficient in JA or ethylene biosynthesis or signaling pathways (*jar1*, *pdf 1.2*, *coi 1*, *ein2*, and *etr1*) showed increased susceptibility [8]. Here, we observed that several genes involved in the ethylene and jasmonate pathways were downregulated under basal conditions, including *ERF1/2*, *JAR1*, and *PDF1.2* (Figure 9A). Moreover, we observed that *LOX2*, the first gene involved in JA biosynthesis, was downregulated at 0 and 40 HPI, which could suggest that *RAC7-OX* plants are deficient in jasmonates. Thus, when methyl jasmonate was applied to *RAC7-OX* plants, we observed a reduction in the lesion area (Figure 10A), confirming that *RAC7* overexpression affects JA biosynthesis.

Furthermore, plants use a wide arsenal of secondary metabolites to fight pathogens. The metabolites derived from indole-3-acetaldoxime such as glucosinolates and phytoalexins, have been shown to be very important against *B. cinerea* infection. The *RAC7-OX* line had downregulated glucosinolate biosynthetic pathways as well as a major part of camalexin biosynthesis, which is shared with indole glucosinolate biosynthesis (Figure 9C). Reduction of these metabolites could also contribute to *RAC7*-induced susceptibility. Interestingly, the indole glucosinolate, camalexin, and JA pathways share several regulatory points and cross-talk [63]. All these results probably point to *RAC7* regulating the upper elements in the signaling transduction, as its overexpression deregulates several pathways simultaneously. This could be the reason why *RAC7* is downregulated on Col-0 when *B. cinerea* and other pathogens infect the plant.

Finally, it has been reported that during the PTI the actin filaments are rapidly reorganized; for instance, in epidermal cells, the actin filaments increase in density in an ADF-dependent manner, but their reorganization can vary depending on the type of pathogen infecting the plant [64]. In the case of fungal pathogens, actin filaments reorganize towards the site of invasion to transport plant defense-related elements, such as antimicrobial substances, to prevent pathogen penetration. However, if the filaments are fragmented and short, plants become more susceptible to pathogens because they cannot transport vesicles with defense elements to the site of invasion and defense genes are not induced [64]. This is the case with the *adf4* mutant, which was impaired in the activation of CDPK-related genes and actin rearrangement in response to bacterial elicitors. Herein, we report a differential cytoskeleton re-arrangement in *RAC7-OX* before *B. cinerea* infection which facilitates the further pathogen colonization (Figure 5). We hypothesize that overexpression of *RAC7* negatively affects actin dynamics, resulting in actin filament architecture that impairs the plant immune response and for the transport of defense elements through actin filaments. This leads to inappropriate signaling for the subsequent activation of defense genes, synthesis of hormones and antifungal compounds, and cell wall reinforcement. With that in mind, we hypothesize that to respond adequately to *B. cinerea*, Col-0 plants show a decrease in *RAC7* transcript and/or a translocation of the protein to the cytosol (where it remains inactive) in the presence of the fungus, thus suppressing the negative regulation exerted by *RAC7* on the activation of the immune system and allowing the plant to activate its defense mechanisms (Figure 11).

## 4. Materials and Methods

### 4.1. Plant Material and Growth Conditions

To analyze the role of *RAC7/ROP9* in the *A. thaliana*–pathogen interaction, two T-DNA insertion lines of *AtRAC7* were used: SALK_019272.56.00 (*RAC7.1-OX*) and SALK_015223.56.00 (*RAC7.2-OX*). Both lines were obtained from the Arabidopsis Biological Resource Center (ABRC). Both lines have a T-DNA insertion in the 5′ UTR region of the *RAC7* gene. A key feature is that these T-DNA lines contain a 35S promoter sequence from the pROK2 plasmid used to generate the insertion lines (Appendix A). Columbia-0 (Col-0) ecotype was used as the wild type plant (WT). *A. thaliana* plants were grown in peat moss plus vermiculite (1:3), in chambers of 50 pots with 1 plant per pot, under greenhouse conditions at 22 °C, 60% humidity and watered once a week for 4 weeks under a long-day photoperiod (16 h light).

### 4.2. Pathogens Infection Assay

*Botrytis cinerea* B05.10 (provided by Brigitte Mauch-Mani, Neuchâtel, Switzerland) was grown on potato dextrose agar (PDA, Sigma-Aldrich, Saint Louis, MO, USA) and spores were harvested on distilled water by scraping the mycelium in a Petri dish as previously described [65]. For inoculation, the spore concentration was adjusted to 5 × 10^4^ spores mL^−1^ in ¼ strength potato dextrose broth (PDB, 6 g/L; Sigma-Aldrich, USA). For analysis of lesion development, a minimum of 3 leaves (per plant) of 4-week-old plants were inoculated by placing 6 µL of the spore suspension on the leaf surface. Inoculated plants were maintained at high humidity, 22 °C in a 24 h dark cycle. The development of lesions was assessed at 40, 48 and 72 h after inoculation. Each infection was performed with a minimum of 20 plants per line and was repeated with at least three independent biological replicates. Lesion development was quantified using Image J analysis software (Fiji Is Just Image J1 http://imagej.net/, accessed on 6 September 2019) [66]. To determine the progression of *B. cinerea* colonization in the Col-0 and *RAC7-OX* plants, and *B. cinerea* gDNA was quantified in infected plants at 6, 16, 40 and 72 HPI, following the methodology proposed by Brouwer [19]. gDNA extracted from uninfected *A. thaliana* plants was used as a negative control. Total gDNA from infected and uninfected plants was extracted with CTAB according to Alonso and Stepanova [67]. In addition, gDNA was extracted from a *B. cinerea* culture, following the same methodology, which was used to perform serial dilutions and elaborate the standard curve (Appendix A). Quantitative real-time PCR (RT-qPCR) analysis was performed using SYBR Green/ROX qPCR Master Mix (2X) (Thermo Scientific™, Waltham, MA, USA) and the specific primers for the *cutinase A* genes GS11F and GS11R (Appendix A). The Ct values were interpolated on the standard curve to determine *B. cinerea* gDNA concentration. The *B. cinerea*-GFP tagged line was previously described [68]. The observation was made by infecting plants as described for *B. cinerea* B05.10. Infected leaves were observed under a confocal microscope using the following conditions (Exitation λ = 488 nm, Emision λ = 520 nm).

For the *Pseudomonas syringae* DC3000 (provided by Ramón Suárez Rodríguez, CEIB-UAEM, México) infection assay, the bacterium was grown in 50 mL of LB medium (Luria Beltrani) supplemented with rifampicin (50 µg mL^−1^) at 28 °C with shaking at 200 rpm until OD_600_ = 0.6–1 was reached. The cells were then harvested by spinning the culture at 2500× *g* for 10 min. The supernatant was discarded, and the cells were resuspended in 10 mM MgCl_2_ to achieve OD_600_ = 0.2. The cell suspension was used to infiltrate 3 leaves per plant of Col-0, *RAC7.1-OX*, and *RAC7.2-OX*. Infiltration was performed with 10 µL of the bacterial suspension and MgCl_2_ as control on the abaxial side of the leaf. From each line, 20 plants were used per biological replicate and the experiment was repeated with three independent biological replicates. Treated plants were covered and kept in the dark for 72 h in a humid chamber at 22 °C. At 72 HPI, the infiltrated leaves were cut and sterilized with 70% ethanol for 1 min, followed by three rinses with sterilized water of 1 min each. The leaves were briefly dried with sterile filter paper and an area of 0.758 cm^2^ per leaf was excised. Leaf discs were macerated in 1 mL of 1 mM MgCl_2_ and serially diluted to 1 × 10^−4^. An amount of 10 µL of each dilution was plated on solid LB plates supplemented with rifampicin (50 µg mL^−1^) and incubated at 28 °C for 48 h. The colonies were then counted and the CFU/cm^2^ per leaf calculated.

For the *Pectobacterium brasiliense* BF45 (kindly donated by Dr Oscar Mascorro from Universidad de Chapingo, El Cooperativo, México) infection assay, bacteria were grown in 50 mL of LB medium at 28 °C with 200 rpm agitation, until OD_600_ = 0.6–1 was reached. Subsequently, the cells were harvested by spinning the culture at 2500× *g* for 10 min. The supernatant was discarded, and the cells were resuspended in LB adjusting the OD_600_ = 0.1. An amount of 6 µL of cell suspension was dropped on 3 leaves per plant of Col-0, *RAC7.1-OX*, and *RAC7.2-OX*. From each line, 20 plants were used per biological replicate, and the experiment was repeated three times. The treated plants were covered and kept in the dark in a humid chamber at 22 °C for 24 h. After the incubation period, the incidence rate (number of infected leaves/total number of leaves) and lesion symptoms (soft rot) were observed.

### 4.3. Quantitative RT-PCR Analysis

For the quantification of the expression of all evaluated genes, leaves of the Col-0 and *RAC7-OX* lines were used as biological material. In all cases and treatments, three biological replicates were used with n = 20 plants per replicate. In the case of pathogen treatment, the methodology described above was followed and tissue was collected at 40 HPI in plants treated with *B. cinerea*, and at 24 HPI in plants treated with *P. brasilense* and *P. syringae*. Total RNA was isolated from leaves using the Plant/Fungi Total RNA Purification Kit according to the manufacturer’s instructions (NORGEN BIOTEK CORP., Thorold, ON, Canada). The purified total RNA was quantified using a UV light spectrophotometer (NanoDrop™, Thermo Scientific™) and the integrity of the purified RNA was assessed using a denaturing agarose gel (1% agarose, 1% chloride). The samples were then treated with DNase I Rnase free (#EN0521, Thermo Scientific™, USA) using 1 µg of total RNA, 1 µL of 10× reaction buffer with MgCl_2_, 1 µL DNAse-free RNAse and DEPC-treated water to a total volume of 20 µL. The samples were incubated at 37 °C for 30 min and the reaction was inactivated by adding 1 µL of 50 mM EDTA and incubating at 65 °C for 10 min. After Dnase treatment, the integrity of the total RNA was checked again using a 1% agarose denaturing gel. Samples with intact RNA were used for cDNA synthesis. For this purpose, 1 µg of total RNA was used, and synthesis was performed using the SCRIPT cDNA Synthesis Kit according to the manufacturer’s instructions (Jena Bioscience, Jena, Germany). The correct synthesis of the cDNAs was verified by endpoint PCR using the actin normalization gene AT3G18780 (Appendix A).

Real-time quantitative PCR (RT-qPCR) analysis was performed using SYBR Green/ROX qPCR Master Mix (2X) (#K0221, Thermo Scientific™, USA), and the specific primers for the genes evaluated are listed in Appendix A. Each PCR reaction (10 µL final volume) contained 4 µL cDNA (1:40), 1 µL mixed primers and 5 µL master mix. The thermal cycler conditions were as follows 2 min at 50 °C, followed by 15 min at 95 °C, then 40 cycles of 15 s at 95 °C and 1 min at 60 °C, with a final dissociation gradient from 60 °C to 95 °C, at a rate of 0.3 °Cs^−1^. RT-qPCR was performed on the StepOnePlus™ Real-Time PCR System (Applied Biosystems™, Waltham, MA, USA). The efficiency of the RT-qPCRs was confirmed using a standard curve for each primer evaluated. The *Actin* (AT3G18780), *ubiquitin* (AT4G05320), and *CF150* (AT1G72150) genes were used as internal controls and normalization genes. Expression was calculated as 2^−ΔCt^ or Log2FC.

### 4.4. Constructs for GFP:RAC7 Translational Fusion and Subcellular Localization in N. benthamiana Leaves

The *RAC7* CDS was obtained by PCR from the previous assembled and sequenced plasmid. The purified PCR was cloned using Gateweay technology in pMBC43. The *Agrobacterium tumefaciens* strain GV3101 was transformed with the construction p35s:GFP:RAC7-pMBC43, and used to agroinfiltrate 4-week-old *N. benthamiana* leaves. The transient transformed *N. benthamiana* leaves were inoculated with *B. cinerea* spores (5 × 10^4^ spores mL^−1^) and kept in humidity chamber for 24 h. The transient transformed *N. benthamiana* leaves, infected and not infected with *B. cinerea*, were inspected by confocal microscopy using the same condition for all the samples (excitation at 488 nm and emission 530 nm). The images were processed using Image J analysis software (Fiji Is Just Image J1).

### 4.5. ROS Detection

Staining with 5-(and 6)-carboxy-2′,7′-dichloro dihydrofluorescein diacetate (DCF-DA), 3,3′-diaminobenzidine (DAB) and nitroblue tetrazolium (NBT) was used to determine ROS accumulation. In all cases, 15 plants per tested line were used and each line was evaluated at 0, 6, 16, and 40 HPI. For the determination of ROS production with DCF-DA, leaves were immersed in 60 µM DCF-DA in a standard medium (1 mM KCl, 1 mM MgCl_2_, 1 mM CaCl_2_, 5 mM 2 mM MES 2-(*N*-morpholino) ethanesulfonic acid pH 6.1). The leaves were then washed and observed under a GFP filter microscope (excitation 480 nm, emission 527 nm). To determine H_2_O_2_ accumulation, the collected leaves were immersed in a solution of DAB (1 mg mL^−1^) and placed in the dark for 24 h. After this time, the solution was removed and a solution of ethanol: acetic acid: glycerol (3:1:1) was added for 24 h. The presence of a brown precipitate, indicating H_2_O_2_ production in the leaves, was then observed for qualitative determination [63]. For the determination of O^2−^ accumulation, the leaves to be evaluated were immersed in a solution of NBT (1 mg mL^−1^) in phosphate buffer pH = 7.5 and left in the dark for 24 h. After this time, the solution was removed and a solution of ethanol:acetic acid:glycerol (3:1:1) was added for 24 h. O^2−^ accumulation is visualized by the appearance of blue spots at the accumulation sites [65].

### 4.6. Determination of Callose Accumulation

To determine whether *RAC7-OX* deposited less or more callose at the pathogen entry site, callose deposition was observed at 0, 6, 16, and 40 HPI. In all cases, 15 plants (one leaf per plant) were used per line tested. Callose accumulation was determined using 0.01% aniline blue. For this purpose, the collected leaves were immersed in an ethanol:acetic acid solution (3:1) for 24 h to distain them. After this time, the solution was removed, and a 0.01% (*w*/*v*) solution of aniline blue was added for 3 h. The leaves were then observed using a Zeiss Axioskop 2 microscope coupled to a Zeiss AxioCam MRc camera. Aniline blue staining uses a fluorochrome that forms a complex with callose for microscopic visualization through an ultraviolet filter. In this way, the deposited callose is observed as yellow-green dots in the plant tissue. An excitation wavelength of 370 nm and an emission wavelength of 509 nm were used for visualization, with an exposure time of 300 ms.

### 4.7. Determination of Leaf Permeability by Chlorophyll Leakage

For the determination of chlorophyll content, one gram of leaves from each tested line was weighed. The leaves were immersed in 80% ethanol and kept in the dark. The absorbance was measured at 664 nm and 647 nm every hour for 5 h and the chlorophyll content was determined using the following equation: Total μmoles of chlorophyll = 7.93(A_664_) + 19.53(A_647_) [69].

### 4.8. Actin Cytoskeleton Staining

The actin cytoskeletons of 4-week-old leaves of Col-0 and *RAC7.2-OX* treated with and without *B. cinerea* (6 HPI) were visualized with epifluorescence microscopy on leaves segments fixed with Alexa-Phalloidin. The samples were excited at 484 nm, and emission was collected at 530 nm (20 nm band pass). All filters used were from Chroma Technology, and image acquisition and analysis were carried out using MetaMorph/MetaFluor software V7.8 (Universal Imaging, Molecular Devices, San Jose, CA, USA).

### 4.9. RNA Extraction, RNA Sequencing, and Analysis

For RNAseq, Col-0 and *RAC7-OX* leaves were sprayed with 5 × 10^4^ spores/mL of *B. cinerea*. A total of 25 whole rosettes per two biological replicates per line were collected at 40 HPI and under non-infection conditions. This value of 40 HPI was selected due to this being the time we check *RAC7* transcript decreases. Total RNA was isolated and purified using the Spectrum™ Total RNA kit (Sigma-Aldrich, USA). The concentration and purity of the extracted RNA was measured using NanoDrop™ 2000 (Thermo Fisher Scientific, Inc., Waltham, MA, USA). Construction and sequencing of the cDNA library was performed by Beijing Genomics Institute (BGI) Americas using DNBSeq™ technology. The sequences are publicly available at the following link: https://dataview.ncbi.nlm.nih.gov/object/PRJNA986958?reviewer=q8elucq0lg3rpdto85gifoomrt (accessed on 21 December 2023). Sequencing was performed using 100 bp paired-end reads. Approximately 20 million reads per sample were aligned to the *A. thaliana* genome (TAIR version 10) using Bowtie2 (v2.3.5) [70]. Calculation of gene expression levels was performed using RSEM (RNA-seq by expectation maximization method) (v1.3.3) (Li and Dewey, 2011 [71]). Differentially expressed genes were identified using DESeq2 software (v 1.22.1) on the IDEAMEX (Integrated Differential Expression Analysis MultiEXperiment) platform [72]. Parameters were adjusted as Log2FC ≥ 1 or ≤−1 and adjusted *p*-value ≤ 0.05. DEGs of *RAC7-OX* vs. Col-0 were determined at 0 and 40 HPI, and the analysis of DEG ontologies (GO) was performed employing PANTHER (v16.0) and the database (DAVID) (v6.8). Bubble plot graphics were performed with the ggplot2 library using RStudio (v1.4.1106). Gene expression analysis of the plant biotic stress pathway was performed with MAPMAN (v 3.7.0).

### 4.10. Phytohormone Chemical Complementation

For chemical complementation, Col-0, *RAC7.1-OX*, and *RAC7.2-OX* were treated by spraying entire rosettes with Methyl Jasmonate (0.2% *v*/*v*), Salicylic Acid (500 µM), or Camalexin (200 µg mL^−1^), dissolved in DMSO (for JA and Camalexin) or ethanol (SA). Mock controls were performed using DMSO or ethanol, respectively. After treatments, the plants were kept in a humid chamber for 24 h, then left to dry for two hours, and infected with *B. cinerea* by dropping 6 µL of 5 × 10^4^ spore/mL on plant leaves. The plants were kept under high humidity and in darkness for 72 h. Lesion development was analyzed using Image J analysis software (Fiji Is Just Image J1) [66]. Each treatment was performed with a minimum of 20 plants and was repeated 3 times.

### 4.11. Statistical Analysis

The graphs were generated using the ggplot2 library with RStudio (v1.4.1106) or GraphPad Prism 9, and all statistical analyses were performed with GraphPad Prism 9. Differences between the means of variables were analyzed using the Kruskal–Wallis test, followed by post hoc Dunn’s multiple comparison test. Analysis of variance (ANOVA) was performed on the comparison between hormone-supplemented, *B. cinerea*-treated and untreated plants, followed by Tukey’s HSD test to separate means. A probability of *p* ≤ 0.05 was considered significant.

## 5. Conclusions

*RAC7* is a negative regulator of the immune response of *A. thaliana* to several pathogens with different lifestyles. Its overexpression in *Arabidopsis* results in transcriptional dysregulation, affecting genes associated with the actin cytoskeleton, plant defense, cell wall status, signal transduction, and the biosynthesis of hormones and secondary metabolites, both before and during *B. cinerea* infection. All this results in the plant immunosuppressive status, not only before *B. cinerea* infection, but also before *P. syringae* and *P. brasiliense*. The fact that *RAC7* overexpression induces immunosuppression against several microorganisms raises the question of whether *RAC7* overexpression could lead to interactions with beneficial microorganisms that do not occur in nature, since beneficial microorganisms are recognized as pathogens in the early stages of colonization and this may determine whether a symbiosis is established or not. However, overexpression of *RAC7* could block this checkpoint and allow unnatural mutualistic interactions to occur, opening a new door for the study of plant immune responses during plant–microbe mutualistic interactions.

## Figures and Tables

**Figure 1 ijms-25-00591-f001:**
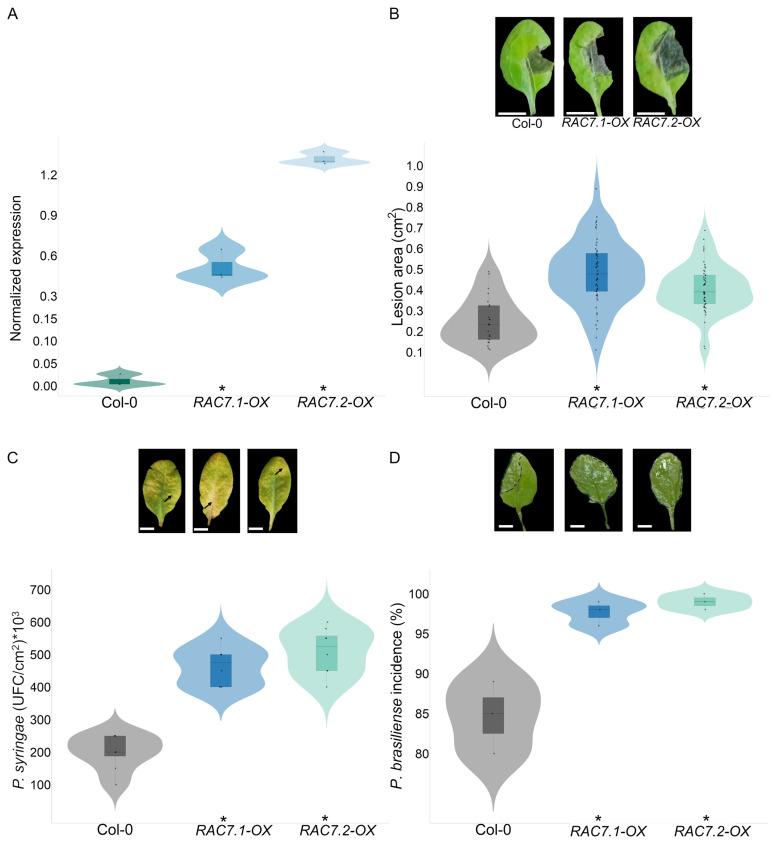
Infection assays were conducted using the *RAC7.1-OX*, *RAC7.2-OX*, and Col-0 lines with pathogens with different lifestyles. (**A**) Analysis of *RAC7* expression in the *RAC7.1-OX*, *RAC7.2-OX*, and Col-0 lines without inoculation. *CF150* (AT1G72150) was used as the normalization gene. (**B**) Leaves of *RAC7.1-OX*, *RAC7.2-OX* and Col-0 inoculated with *B. cinerea*, the lesion area was calculated 72 HPI. (**C**) Leaves of *RAC7.1-OX*, *RAC7.2-OX*, and Col-0 inoculated with *P. syringae*, UFC/cm^2^ was calculated as 72 HPI. (**D**) Leaves of *RAC7.1-OX*, *RAC7.2-OX*, and Col-0 inoculated with *P. brasiliensis*, the incidence rate was calculated as 72 HPI. In the boxplots, the center line represents means values of three independent experiments (n > 20); box limits, upper and lower quartiles; whiskers, 1.5× interquartile range; the points represent individual data points. The asterisk indicates statistical significance between the RAC7-OX lines and Col-0 according to a one-way ANOVA test and a post hoc analysis (*p* < 0.05). Pictures are representative images of lesion development. Scale = 1 cm (**B**–**D**). Arrows indicate yellowing lesions caused by *P. syringae* and dashed lines indicate lesions caused by *P. brasilense* infection.

**Figure 2 ijms-25-00591-f002:**
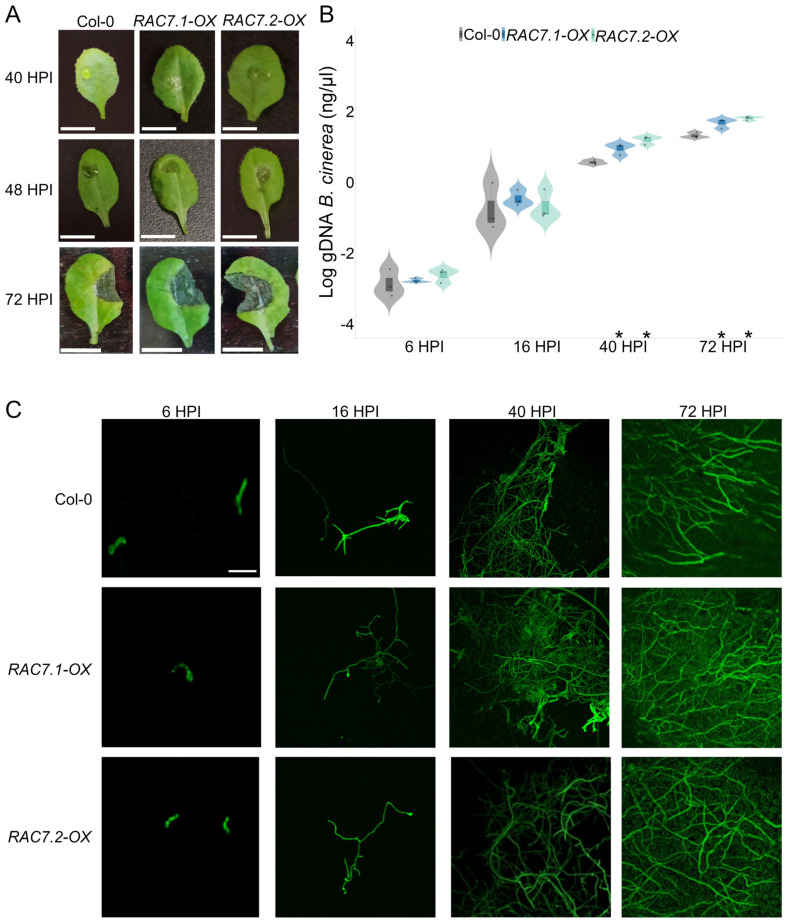
*RAC7* overexpression increases the amount of *B. cinerea* infecting the leaves. (**A**) Lesion development at 40, 48 and 72 HPI in *RAC7.1-OX*, *RAC7.2-OX*, and Col-0. Quantification of *B. cinerea* gDNA in leaves of *A. thaliana* Col-0 and *RAC7-OX* lines at 6, 16, 40 and 72 HPI. (**B**) In the boxplots, the center line represents means values of three independent experiments (n > 20); box limits, upper and lower quartiles; whiskers, 1.5× interquartile range; the points represent individual data points. The asterisk indicates statistical significance between the *RAC7* overexpressors and Col-0 according to a one-way ANOVA test and a post hoc analysis (*p* < 0.05). (**C**) Growing of *B. cinerea* tagged with GFP in *RAC7.1-OX*, *RAC7.2-OX*, and Col-0 at 6, 16, 40 and 72 HPI. Scale = 1 cm (**A**), 50 µm (**C**).

**Figure 3 ijms-25-00591-f003:**
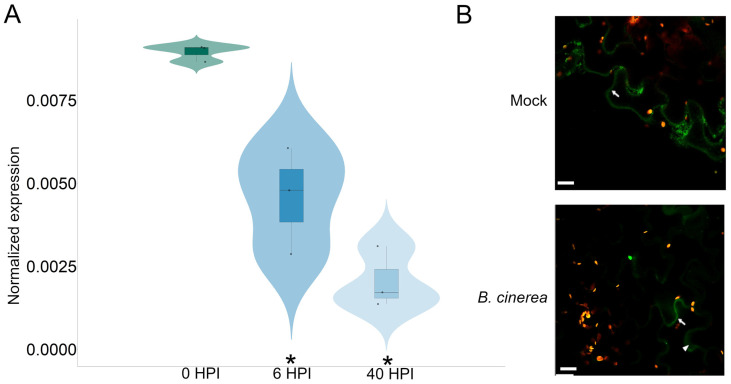
*RAC7* behavior in response to *B. cinerea* infection. (**A**) Transcript expression profile in Col-0 leaves infected with *B. cinerea* at 0, 6, and 40 HPI. Normalized expression was calculated using 2^−ΔCt^ and normalized with respect to *CF150*. The box represents mean values (±SD) of three independent experiments (n > 20). The asterisk indicates statistical significance between the 6 and 40 HPI with respect to 0 HPI, according to a one-way ANOVA test and a post hoc analysis (*p* < 0.05). (**B**) Subcellular location of *RAC7* (35S::GFP:RAC7) in *N. benthamiana* leaves with and without *B. cinerea* inoculation (24 HPI). The arrows indicate the cytoplasmic membrane and the arrowheads the cytoplasm. Scale = 20 µm.

**Figure 4 ijms-25-00591-f004:**
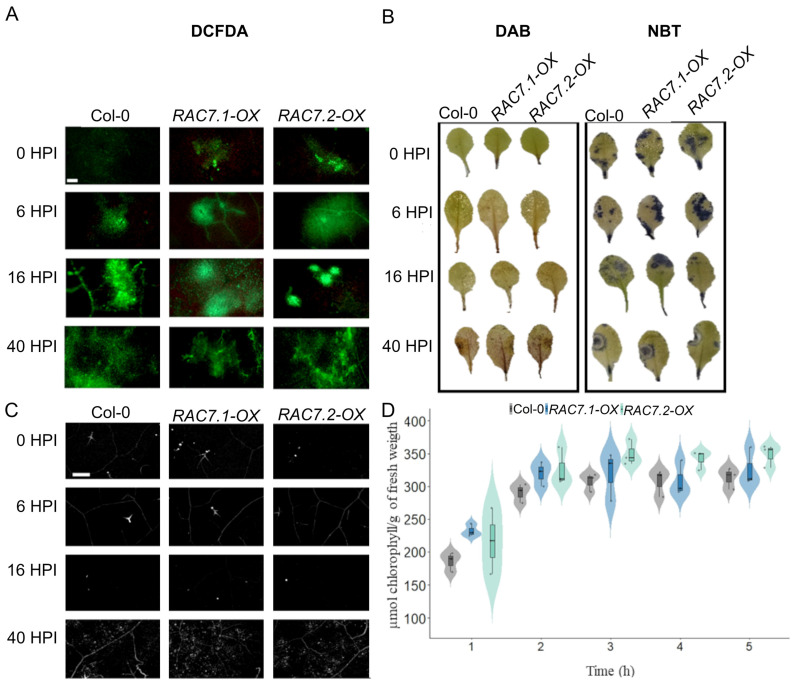
ROS production, callose deposition, and leaves permeability are not disturbed in *RAC7-OX*. (**A**) Accumulation of ROS (measured with DFCDA), (**B**) H_2_O_2_ (measured with DAB) and O^2−^ (measured with NBT) in Col-0, *RAC7.1-OX*, and *RAC7.2-OX* plants at 0, 6, 16, and 40 HPI. (**C**) Callose deposition in Col-0 and *RAC7* overexpressing lines at 0, 6, 16, and 40 HPI. Each photograph represents the consensus of 15 plants per line. (**D**) Chlorophyll leaching 1, 2, 3, 4, and 5 h after immersion of the leaves on 80% EtOH. In the boxplots, the center line represents means values of three independent experiments (n > 20); box limits, upper and lower quartiles; whiskers, 1.5× interquartile range; the points represent individual data points. Scale = 200 μm (**A**,**C**).

**Figure 5 ijms-25-00591-f005:**
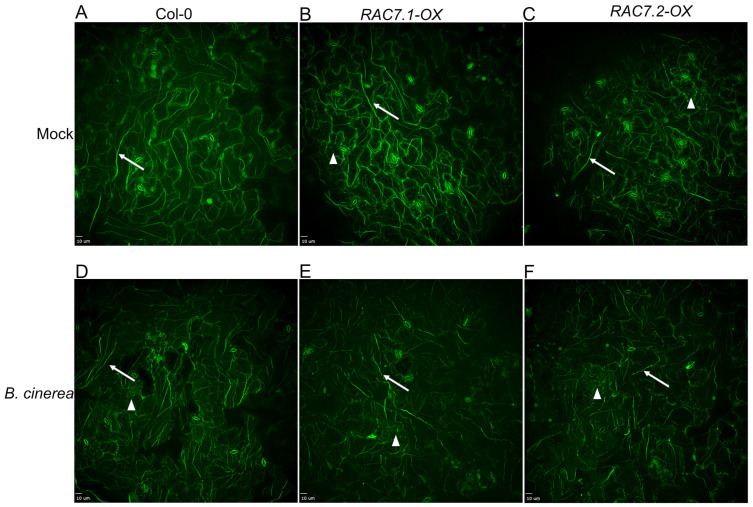
Actin cytoskeleton organization in epidermic cells of Col-0 and *RAC7-OX.* (**A**,**D**) Imaging of the actin microfilament organization in Col-0, (**B**,**E**) *RAC7.1-OX*, and *RAC7.2-OX (***C**,**F**), without (**A**–**C**) and with *B. cinerea* at 6 HPI (**D**–**F**). The arrows point to long filaments and arrowheads indicate fragmented filaments. Scale 10 μm.

**Figure 6 ijms-25-00591-f006:**
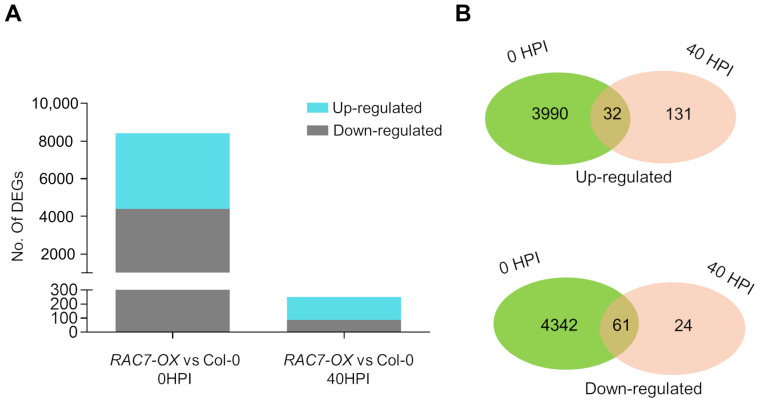
Transcriptome analysis of *RAC7-OX* with respect to Col-0 at basal conditions and 40 HPI. (**A**) Number of differentially expressed genes (DEGs) (Log2FC ≥ 1 or ≤–1, *p*-value ≤ 0.05) on *RAC7-OX* with respect Col-0 at 0 and 40 HPI. (**B**) The Venn diagrams show the overlap of times in upregulated and downregulated genes in *RAC7-OX*.

**Figure 7 ijms-25-00591-f007:**
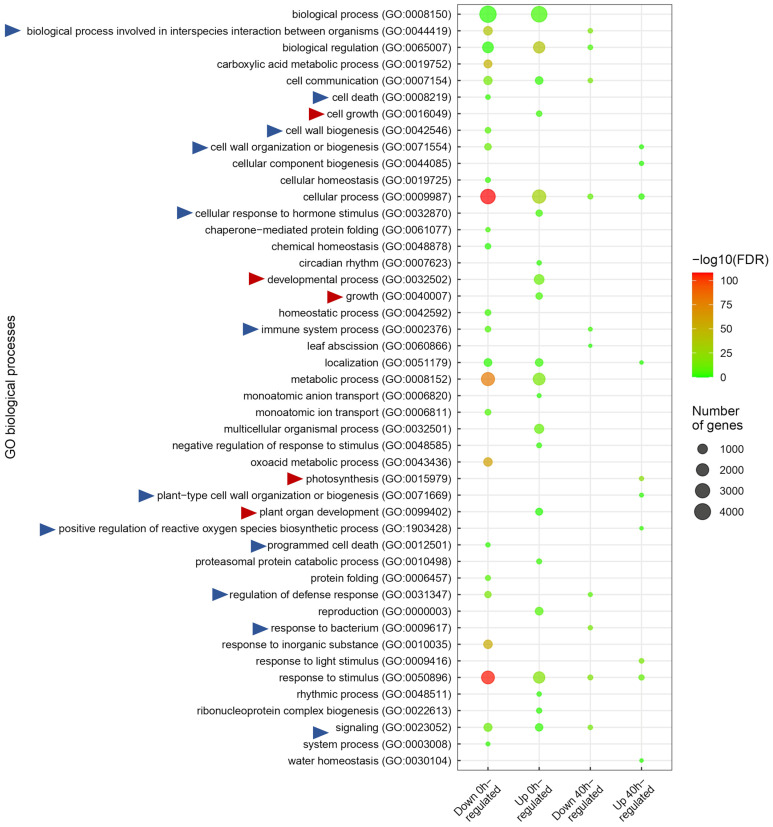
Gene ontology (GO) enrichment of *RAC7-OX* vs. Col-0 DEGs at 0 and 40 HPI. The bubble color indicates the significance of the term (−log10 FDR), and the size indicates the number of genes associated with each term. The blue and red triangles indicate GO related to plant defense against pathogens and development, respectively.

**Figure 8 ijms-25-00591-f008:**
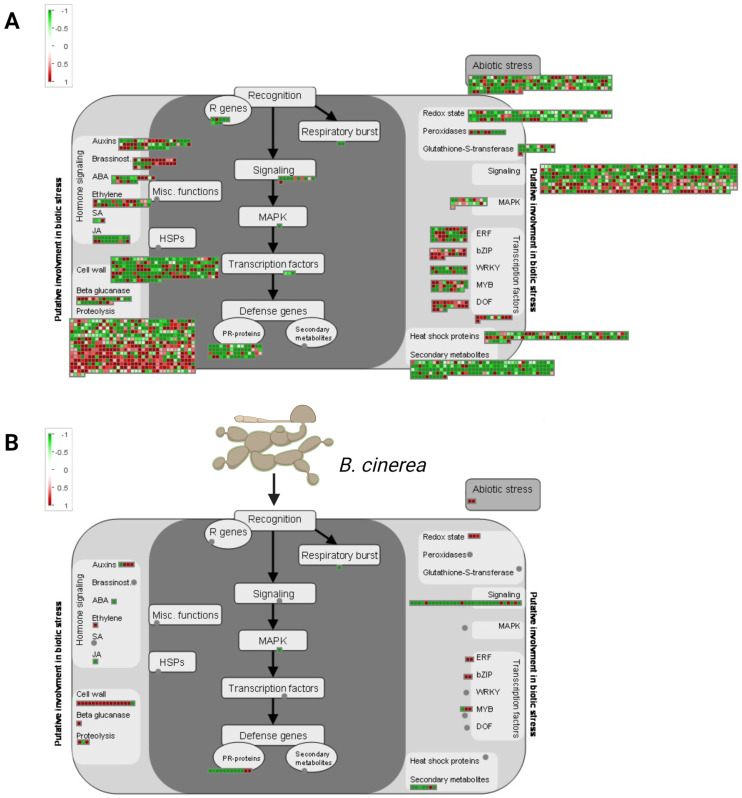
MapMan visualization of *RAC7-OX* DEGs in the plant biotic stress category at (**A**) 0, and (**B**) 40 HPI. The squares represent significantly differentially expressed *Arabidopsis* TAIR10 gene models (padj < 0.05) from the corresponding categories, in the indicated conditions. Log2 fold changes are indicated as a gradient between red (1) and green (−1). The grey dot represents the lack of significantly differentially expressed genes in the corresponding category.

**Figure 9 ijms-25-00591-f009:**
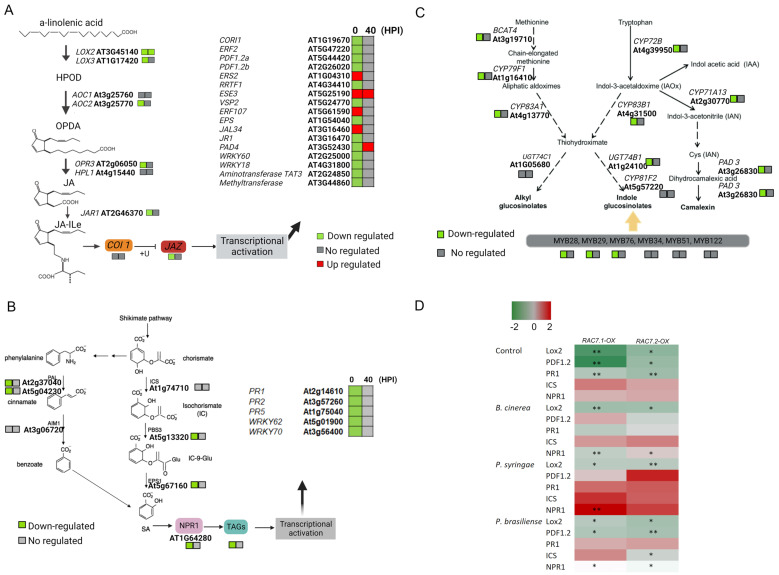
Hormonal, glucosinolates and camalexin biosynthesis is downregulated in *RAC7-OX*. (**A**) Jasmonic acid, (**B**) Salicylic acid, (**C**) Glucosinolates and Camalexin biosynthesis pathways, signal transduction, and induced genes. The squares represent 0 HPI and 40 HPI. Log2 fold changes are indicated as a gradient between red (1) and green (−1). Grey represents the lack of significantly differentially expressed genes in the corresponding category. (**D**) Log_2_FC of *LOX2*, *PDF1.2*, *PR1*, *ICS1*, *NPR1* at basal conditions and treated with *B. cinerea* (40 HPI), *P. syringae* (24 HPI), and *P. brasilense* (24 HPI). The asterisks indicate significant difference: * *p* < 0.05; ** *p* < 0.01.

**Figure 10 ijms-25-00591-f010:**
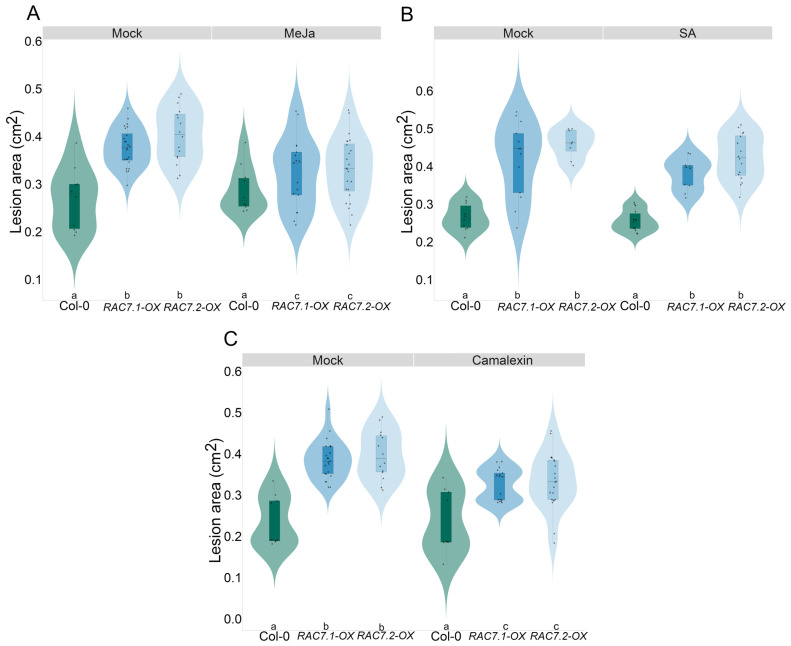
Chemical complementation of Col-0, *RAC7.1-OX*, and *RAC7.2-OX*. Lesion area of plants pre-treated with (**A**) Methyl Jasmonate, (**B**) Salicylic acid, (**C**) and Camalexin, and infected with 5 × 10^4^ spore/mL of *B. cinerea*. In the boxplots, the center line represents means values of three independent experiments (n > 20); box limits, upper and lower quartiles; whiskers, 1.5× interquartile range; points represent individual data points. Letters indicate statistical differences according to Tuky’s test.

**Figure 11 ijms-25-00591-f011:**
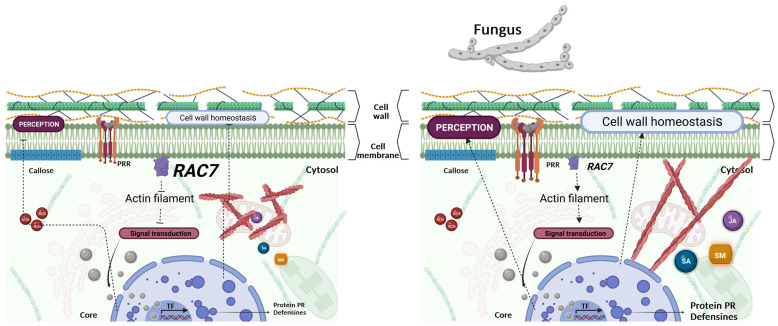
Schematic representation illustrating how the *RAC7* would potentially negatively regulate plant immunity in *A. thaliana* by modulating the actin cytoskeleton and Ca^++^ signaling. In the Col-0 plants, in no challenge conditions, the level of *RAC7* expression keeps suppressed process related to plant defense such as pathogen perception, cell wall homeostasis, signal transduction, JA and SA induction, and antimicrobial secondary metabolism (SM) biosynthesis. Once a pathogen infects the plant, *RAC7* expression is suppressed and the inhibition is blocked, resulting in immunity activation. Created with BioRender.com.

## Data Availability

The data are available at https://dataview.ncbi.nlm.nih.gov/object/PRJNA986958?reviewer=q8elucq0lg3rpdto85gifoomrt (accessed on 21 December 2023).

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
