# Peer review of "AtRAC7/ROP9 Small GTPase Regulates A. thaliana Immune Systems in Response to B. cinerea Infection"

_ijms, 2024, doi:10.3390/ijms25010591_

Round 1

Reviewer 1 Report

Comments and Suggestions for Authors

Garcia-Soto and colleagues conducted a study on RAC7 Arabidopsis overexpression lines that were infected with Botrytis cinerea or challenged with bacteria (Pseudomonas syringae and Pectobacterium brasiliense). The lines exhibited higher susceptibility to both the fungus and the bacteria. Transcriptional down-regulation of defense-related pathways before and during the fungal infection were observed, and the authors suggest a negative role of the RAC7 small GTPase during the infection process. The topic is of interest to the readership of IJMS. However, before the manuscript can be published, a thorough revision is necessary.

It is known that pleiotropic effects may occure in overexpression lines, leading to diverse outcomes. The authors do not include both lines in all of their studies. To accept the paper, data should be provided for actin cytoskeleton architecture in line RAC7.1-OX. Furthermore, the transcriptom analysis results by RNAseq should be verified by qPCR for both lines for selected genes.

Results: The results section could be condensed further. There are text elements that belong in the Materials and Methods section or in the Discussion. For example line 108-line113: To investigate the possible role of RAC7, we analyzed two RAC7 T DNA insertion lines (SALK_019272.56.00.x, RAC7.1 OX ; SALK_015223.56.00.x, RAC7.2 OX ). Both lines have a T DNA insertion in the 5' UTR region of the RAC7 gene…this belongs to the Materials and Methods part.

Figure 1A: It was normalized to what in this case?

Figure 1B, Pls change cm-> cm2

Figure 2C, Pls change 48HPI -> 40 HPI

Figure 3A: It is not clear to me to which of the referenced genes it was normalized.

Figure 5: Why only results from line RAC7.2-OX are shown? According to possible pleiotropic effects both overexpression lines should be tested. This is the same for data shown in Figure 6. 

Materials and Methods

Line 618/620: Please insert the authors name there.

Line 658: In all cases and treatments, three biological replicates were used with n=20 plants per replicate. Do the auhtors mean three independent experiments? Not clear to me, please clarify.

Line 681: 2 min at 50°C, followed by 15 min at 95°C, then 40 denaturation cycles of 15 s at 95°C, 30 s annealing at 60°C and 30 s extension at 72°C… Is this correct?

Line 694: spores (5*10 4 spores/mL) -> Pls better use spores mL-1 for concentrations… see also other cases in the text.

Line 745: Why do the authors chose 40 HPI for RNAseq analysis?

Line 779: …on chemical treatment. What about on infection/inoculation? Please clarify.

Conclusions: The last two sentence line 789-791 are not fully clear to me how the related the study. Maybe the authors could drop a few more words on this.

References: In the reference list, there are some errors, for example, species names were not italicized. Please check and correct.

Author Response

Thank you very much for taking the time to review this manuscript, your comments have been very timely and will contribute to improving this work. Below you will find the detailed responses and the corresponding revisions and corrections highlighted in yellow.

The responses point by point are included in the attached file. 

Reviewer 2 Report

Comments and Suggestions for Authors

Dear Authors

The present manuscript entitled “AtRAC7/ROP9 small GTPase regulates A. thaliana immune systems front B. cinerea infection” presented an investigation regarding the potential role of the Arabidopsis RAC7 gene during infection with B. cinerea. Findings on pathway analysis of differentially expressed genes suggested that RAC7 influences pathogen perception, cell wall homeostasis, signal transduction, biosynthesis and response to hormones and antimicrobial compounds through actin filament modulation. The authors claimed the negative role of RAC7 small GTPase during A. thaliana-B. cinerea interaction for the first time. The methodology is well designed and results were nicely presented; although there are certain opportunities for further improvement, please find below.

1.      Line 599-605, Please include more details of plant growth conditions, what kind of pots were used in greenhouse condition, how many plants per pots, number of biological and technical replicates etc.

2.      Source of Botrytis cinerea B05.10 culture?

3.      How the spores were harvested, please include the details.

4.      How the plants were inoculated? by pinching the leaves or on intact leaves?

5.      Similarly the source of Pseudomonas syringae DC3000?

6.      Pectobacterium brasiliense BF45 source?

 Thank you

Author Response

(The authors gave the same response as above.)
